# CPT1A Over-Expression Increases Reactive Oxygen Species in the Mitochondria and Promotes Antioxidant Defenses in Prostate Cancer

**DOI:** 10.3390/cancers12113431

**Published:** 2020-11-18

**Authors:** Molishree Joshi, Jihye Kim, Angelo D’Alessandro, Emily Monk, Kimberley Bruce, Hanan Elajaili, Eva Nozik-Grayck, Andrew Goodspeed, James C. Costello, Isabel R. Schlaepfer

**Affiliations:** 1Department of Pharmacology, University of Colorado Anschutz Medical Campus, Aurora, CO 80045, USA; Molishree.Joshi@cuanschutz.edu (M.J.); Andrew.Goodspeed@cuanschutz.edu (A.G.); James.Costello@cuanschutz.edu (J.C.C.); 2Division of Medical Oncology, University of Colorado Anschutz Medical Campus, Aurora, CO 80045, USA; Jihye.Kim@cuanschutz.edu (J.K.); Emily.Monk@cuanschutz.edu (E.M.); 3Department of Biochemistry, University of Colorado Anschutz Medical Campus, Aurora, CO 80045, USA; Angelo.Dalessandro@cuanschutz.edu; 4University of Colorado Comprehensive Cancer Center, Anschutz Medical Campus, Aurora, CO 80045 USA; 5Division of Endocrinology, Metabolism and Diabetes, University of Colorado Anschutz Medical Campus, Aurora, CO 80045, USA; Kimberley.Bruce@cuanschutz.edu; 6Department of Pediatrics Critical Care, University of Colorado Anschutz Medical Campus, Aurora, CO 80045, USA; Hanan.Elajaili@cuanschutz.edu (H.E.); Eva.Nozik@cuanschutz.edu (E.N.-G.)

**Keywords:** CPT1A, prostate cancer, fatty acids, serine, androgen response, ROS, oxidative stress

## Abstract

**Simple Summary:**

Prostate cancer (PCa) is the most common cancer in men and the second highest contributor to cancer deaths. Targeting lipid catabolism enzymes in PCa may offer new avenues for therapeutic approaches. During the last decade, carnitine palmitoyl transferase I (CPT1A) has been identified as a potential therapeutic target for a growing list of cancers. In this study, we have tested the hypothesis that excess CPT1A plays a key role in supporting adaptation to stress and antioxidant defense production in PCa cells. Specifically, we have studied molecular differences between CPT1A gain and loss of function models, revealing genetic and metabolic vulnerabilities that could be targeted to avoid progression to neuroendocrine differentiation, a lethal form of the disease. Examining public datasets, we have also found that excess CPT1A expression leads to worse progression-free survival in PCa patients.

**Abstract:**

Cancers reprogram their metabolism to adapt to environmental changes. In this study, we examined the consequences of altered expression of the mitochondrial enzyme carnitine palmitoyl transferase I (CPT1A) in prostate cancer (PCa) cell models. Using transcriptomic and metabolomic analyses, we compared LNCaP-C4-2 cell lines with depleted (knockdown (KD)) or increased (overexpression (OE)) CPT1A expression. Mitochondrial reactive oxygen species (ROS) were also measured. Transcriptomic analysis identified ER stress, serine biosynthesis and lipid catabolism as significantly upregulated pathways in the OE versus KD cells. On the other hand, androgen response was significantly downregulated in OE cells. These changes associated with increased acyl-carnitines, serine synthesis and glutathione precursors in OE cells. Unexpectedly, OE cells showed increased mitochondrial ROS but when challenged with fatty acids and no androgens, the Superoxide dismutase 2 (SOD2) enzyme increased in the OE cells, suggesting better antioxidant defenses with excess CPT1A expression. Public databases also showed decreased androgen response correlation with increased serine-related metabolism in advanced PCa. Lastly, worse progression free survival was observed with increased lipid catabolism and decreased androgen response. Excess CPT1A is associated with a ROS-mediated stress phenotype that can support PCa disease progression. This study provides a rationale for targeting lipid catabolic pathways for therapy in hormonal cancers.

## 1. Introduction

Prostate cancer (PCa) is the most common cancer in men and the second highest contributor to cancer deaths [1]. Although initially highly effective as a treatment for metastatic PCa, androgen deprivation therapy is characterized by a predictable emergence of resistance, a disease state termed castration-resistant prostate cancer (CRPC) [2]. An important feature of CRPC is the reactivation of androgen receptor (AR) signaling, an event reflected by progressive rises in serum prostate-specific antigen (PSA), a gene product regulated by the androgen receptor (AR) [3]. Substantial evidence has documented that the majority of AR-regulated genes (androgen-response hallmark genes) are re-expressed in most CRPCs, and several mechanisms capable of maintaining AR activity have been established [4,5]. Alternatively, disease progression can arise in the absence of a functional AR as an extremely aggressive and metastatic variant called small cell or neuroendocrine PCa [6,7]. Although genetic alterations are known to promote this aggressive state of the disease [8], we currently lack mechanistic insight as to which metabolic pathways play important roles in PCa plasticity, resistance and progression to lethal disease.

Mitochondrial fatty acid β-oxidation is the major pathway for the catabolism of fatty acids, and it plays an essential role in maintaining whole body energy homeostasis. The transfer of fatty acids into the mitochondria for oxidation happens in an organized, controlled way. The enzyme carnitine palmitoyl transferase I (CPT1A) resides in the outer mitochondrial membrane and catalyzes the reversible transfer of acyl groups between coenzyme A (CoA) and L-carnitine, converting acyl-CoA esters into acyl-carnitine esters. These acyl-carnitines can then enter the matrix where β-oxidation takes place [9]. The activity of CPT1A produces a greater number of acetyl groups, which can be used for energy, de novo lipid synthesis and acetylation reactions, including histone acetylation in the nucleus [10].

Support for a strong association between specific dietary lipids and PCa is still unclear. For example, a recent paper showed that a high fat diet promotes metastatic prostate cancer [11]. Although this is a great step forward in understanding the clinical role of dietary lipids promoting metastatic PCa, it does not address how tumor cells use the lipids to their benefit. Particularly, it is not clear how CPT1A activity could promote a metabolic environment conducive to cell transformation, cancer progression and drug resistance [12].

Targeting lipid catabolism enzymes in PCa may offer new avenues for therapeutic approaches [13]. During the last decade, CPT1A has been identified as a potential therapeutic target for a growing list of cancers [14]. In these cancers, CPT1A expression is increased, and/or its inhibition has been reported to have antitumor effects. Our group has shown that fatty acid oxidation is an important pathway in cancer metabolism in PCa cells when using the CPT1 inhibitor etomoxir [15,16,17]. The blockade of CPT1A with etomoxir produced unresolved endoplasmic reticulum (ER) stress leading to cell apoptosis and decreased tumor growth in vivo. Indeed, lipid oxidation in the mitochondria of aggressive cancer cells can provide adaptation to stress by promoting the generation of antioxidant molecules, like Nicotinamide adenine dinucleotide phosphate (NADPH) [18].

Loss of balance between the antioxidant defense and oxidant production in the cells, which commonly occurs as a secondary feature in many diseases, is loosely termed “oxidative stress”. This balance is important because the intracellular redox environment must be more reducing than oxidizing to maintain optimal cell function. Oxidative stress and ER stress are linked to multiple pathologies, including metabolic, neurodegenerative, immune, and neoplastic diseases. Studies on these two cellular stresses have not only contributed to our understanding of disease, but also opened new avenues to next-generation therapies for these illnesses [19,20]. Their exact role in supporting cancer survival and drug resistance remains unknown.

Since oxidative stress is usually coupled with ER stress [21], the unfolded protein response (UPR) has evolved to handle both proteins folding defects and oxidative challenge. In fact, excess nutrients are proposed to induce ER stress and oxidative stress in pancreatic beta-cells, including glucotoxicity and lipotoxicity. For example, chronic excess glucose induces the pro-apoptotic UPR and oxidative stress in beta-cells cells in vitro and in mice models [22]. In addition, free fatty acids, such as palmitate, induce ER stress and oxidative stress and cause apoptosis in β cells [23]. Cancer cells, especially hormone-dependent cancers, like PCa, exploit these stress responses to deal with the oncogenic and metabolic reprograming, making them more adaptable to affront the insults from drug cancer therapies. Indeed, conditional knockout of the ER chaperone GRP78 (HSPA5) in the prostate of mice with PTEN inactivation suppressed prostate cancer growth [24]. The role of CPT1A in supporting adaptive ER stress responses and redox regulation has not been studied.

In this study, we have tested the hypothesis that excess CPT1A plays a key role in supporting adaptation to stress and antioxidant defense production in CRPC cells. We have used the LNCap-C4-2 model as it represents advanced CRPC that can grow in androgen depleted conditions and form metastasis [25]. We have also explored publicly available clinical databases to support our findings. Specifically, we have studied molecular differences between CPT1A gain and loss of function C4-2 models, revealing metabolic vulnerabilities that could be targeted to avoid progression to neuroendocrine differentiation.

## 2. Results

### 2.1. Cells with Overexpression of CPT1A Show a Lipid Catabolic Phenotype and Increased Growth When Supplemented with Fatty Acids

CPT1A fuels lipid beta-oxidation in PCa cells by producing acyl-carnitines in the mitochondria [9]. To characterize the biochemical changes in the newly described knockdown (KD) and overexpression (OE) C4-2 models [10], we used lipid-based assays and metabolomics. Figure 1A shows that CPT1A is significantly increased in the OE cells and this is associated with a significant increase in intracellular lipase activity, a step which liberates fatty acids from triglyceride stores which can then be used for beta oxidation [26] (Figure 1B). Metabolomic analysis of the OE versus KD C4-2 cells showed significant increases in most of the acyl-carnitine identified, from the short and medium-chain (Figure 1C) to the long-chain species (Figure 1D). Furthermore, the KD cells (blue bars) showed decreased production of the abundant long-chain C16:0 (palmitic) and C18:1 (Oleic) species as expected from the decreased activity of CPT1A in these cells. The use of the CPT1 inhibitor etomoxir, also resulted in decreased production of acyl carnitines in C4-2 parental cells treated with the inhibitor for 48 h (Figure 1E). We next studied the effects of fatty acid availability on the colony growth of OE and KD cells (Figure 1F–I). Fatty acids were conjugated to bovine serum albumin (BSA) and growth was normalized to BSA-only treatment for each cell line. Dodecanoate (C12) treatment showed increased growth in OE compared to empty virus (EV) control cells (Figure 1G), and this effect was not observed in the KD cells (Figure 1F). Similar results were obtained with C16:0 supplementation, where OE cells increased in growth (Figure 1I) but not in KD cells (Figure 1H). The addition of other medium- and long-chain fatty acids also produced significant increases in growth in OE cells compared to controls (Appendix A), except for C18:1, which promoted growth in both KD and EV cells. These results may reflect the lipid droplet-inducing effects of oleic acid supplementation in cancer cells, promoting growth and survival [27,28].

### 2.2. RNAseq Analysis of CPT1A OE Cells Shows Increased ER Stress Response, Serine Metabolism and Less AR Signaling

To identify the molecular signatures associated with CPT1A expression, we performed RNAseq in the C4-2 KD and OE cells. Figure 2 and Appendix A show the results of our analysis comparing the OE cells to the KD cells. Briefly, we normalized each cell line to its respective control (Figure 2A), and then we used limma to compare the gene expression differences between OE and KD [29]. Overall, we found 1157 genes upregulated and 1385 genes downregulated when comparing the OE cells to the KD cells. Genes were filtered by an adjusted *p*-value of 0.0001. Using gene set enrichment analysis (GSEA), we found that ER stress and UPR response were some of the most significant pathways increased in the OE cells, pointing to possible lipid-mediated oxidative stress (Figure 2B). Cell division, mitosis and E2F targets pathways were significantly decreased in OE cells, which is in agreement with their decreased growth rate compared to their control line [10]. The androgen response hallmark gene set was significantly decreased in the OE cells and it reflects less dependency of OE cells on AR signaling, which is something we have observed and reported previously [10]. As expected from our metabolic observations, the lipid catabolic process pathway, including the CPT1A gene, was significantly increased in OE cells compared to KD cells. Figure 2C shows a heatmap of genes from the leading-edge analysis in Figure 2B associated with lipid catabolism, response to cellular stress and androgen response pathways. Enrichment plots for these pathways are shown in Appendix A. Examination of the significant genes in these pathways (Figure 2D–F), showed opposite directions of change between OE and KD cells. For most of the genes, the KD cells showed decreases in gene expression when compared to OE cells.

### 2.3. Excess CPT1A Is Associated with Serine and Glycine Metabolism and Glutathione Homeostasis Metabolites

To understand the type of stress imposed in the CPT1A OE cells, we performed global, non-targeted metabolomics in the four cells lines: non-targeting (NT), KD, EV, and OE cells. In parallel to the RNAseq analysis, we focused on the metabolic pathway differences between OE and KD cells after controlling for their respective control lines (EV and NT respectively). Analysis of the top 25 significant pathways revealed that lipid metabolism, glycine and serine metabolism, and glutathione homeostasis were the most significantly changed pathways (Figure 3A and Appendix A). Glycolysis was also significantly different across groups, with the OE cells showing less glycolysis compared to KD cells (Appendix A).

Considering the RNAseq data, we next analyzed the metabolites corresponding to the serine/glycine metabolism. The serine, glycine and one carbon metabolism pathway are a metabolic network upregulated in tumors and of high clinical relevance [30,31]. Except glucose, individual metabolites of the serine/glycine pathway were not changed significantly in OE versus KD (Figure 3B). However, the OE cells showed more intracellular glucose being shunted towards de novo serine biosynthesis compared to control EV cells (Appendix A). In fact, less glucose seemed to be used for glycolysis in the OE cells. These results correlated with significantly increased expression of key serine/glycine pathway genes: D-3-phosphoglycerate dehydrogenase (PHGDH), phosphohydroxythreonine aminotransferase (PSAT1), and Serine hydroxymethyltransferase (SHMT2) (Figure 2E and Figure 3B). SHMT2 is a mitochondrial enzyme that converts serine (3 carbons) into glycine (2 carbons), transferring one carbon to tetrahydrofolate (mitochondrial folate cycle). The increased dimethylglycine and cystathionine levels in OE cells compared to KD cells supports a higher folate cycle activity in these cells (Figure 3B,C).

Metabolites involved in Glutathione (GSH) homeostasis were also significantly changed in the OE cells compared to the KD cells (Figure 3C), suggesting better antioxidant defense. Particularly, cystathionine and cysteine were significantly increased in OE versus KD cells, and between OE cells and controls, suggesting increased cysteine synthesis and availability to generate glutathione. These increases are associated with increased expression of CTH, a cystathionine gamma lyase that breaks down cystathionine to generate cysteine [32], Figure 3D. Unexpectedly, we also observed increased generation of two breakdown products of glutathione via the ChaC Glutathione Specific Gamma-Glutamylcyclotransferase 1 (CHAC1) enzyme. These enzymes are gamma-glutamyl cyclotransferases, which are induced by ER stress and have specific activity towards glutathione [33]. CHAC1 has been shown to break glutathione into 5-oxoproline and cysteinyl-glycine (Cys-Gly), promoting the depletion of glutathione and stress-induced apoptosis in cysteine deprived cancer cells [34]. However, we did not observe a depletion of glutathione in OE cells with higher CHAC1 expression (Figure 3C), suggesting that metabolite recycling mechanisms or compensatory increased production of glutathione may exist in OE cells [35]. Another indication that OE may have more antioxidant defense than KD cells comes from the increased ratio of ascorbate to dehydroascorbate, which is likely maintained by the dehydroascorbate activity of Glutathione S-Transferase Omega 2 (GSTO2) [36], which is significantly upregulated in the OE cells and can recycle the glutathione to a reduced state, Figure 3C,D.

### 2.4. Mitochondrial Reactive Oxygen Species (ROS) Are Increased in CPT1A-OE Cells

Since GSH homeostasis was significantly changed in OE cells, we reasoned that mitochondrial ROS production could be altered in response to increased CPT1A expression. Thus, we next studied the amount of ROS produced in the mitochondria of the cell models. Figure 4A shows the stacked traces of the Electron Paramagnetic Resonance (EPR) assay, highlighting the increased amplitude of the signal in the OE cells compared to all the other samples. Quantification of the signals is shown in Figure 4B, with the OE cells having a 4-fold increase in mitochondrial ROS production compared to KD cells. Examination of the RNAseq data indicated that SOD2, the mitochondrial superoxide dismutase, was significantly decreased in OE cells compared to KD cells (1.4-fold less *p* < 0.0007, Figure 4C). The cytosolic counterpart SOD1 was not significantly changed between OE and KD cells, while the extracellular SOD3 mRNA was significantly increased in the KD compared to the OE cells (*p* = 0.01). Thus, less SOD2 expression in the OE cells likely accounted for the increase in mitochondrial ROS in these cells. Since the OE cells did not show signs of apoptosis induced by excess ROS we next challenged the cells with long chain lipids (oleic and palmitate mixture at 25 μM each) in the presence or absence of androgens, using regular fetal bovine serum (FBS) or charcoal stripped serum (CSS) devoid of steroid hormones. This fatty acid mixture was used because it represents the most common fatty acids circulating in blood and they are substrates for CPT1A activity (Figure 1D). Figure 4D,E and Appendix A, show that SOD2 did not change in response to the lipid stimulation (Figure 4D), but it decreased in the KD cells compared to the OE cells with androgen withdraw conditions (Figure 4E). Thus, in the presence of lipids and androgen deprivation conditions, CPT1A OE cells are likely to cope with the excess mitochondrial ROS from fat oxidation (Figure 4F).

### 2.5. Lipid and Serine Metabolism Genes Are Associated with Less Androgen Signaling and a More Neuroendocrine Phenotype

Gene set enrichment analysis of our RNAseq showed decreased expression of androgen response genes CPT1A OE cells (Figure 5A). Since decreased AR signaling is associated with changes to a neuroendoendocrine phenotype [7], we next looked for markers of neuronal-like differentiation and identified Enolase 2 (ENO2), Synaptophysin (SYP), Neural Cell Adhesion Molecule 2 (NCAM2), and Neurexophilin 4 (NXPH4) genes significantly upregulated in OE versus KD cells. The ENO2 and SYP genes are markers associated with neuroendocrine PCa (NEPC). To investigate the possibility that CPT1A overexpression (OE) is associated with more aggressive disease, we searched previous transcriptome analysis in the public databases. Particularly, we searched for studies that compared adenocarcinoma with aggressive disease like small cell NEPC and focused on serine and one carbon metabolism, CPT1A and AR expression in clinical data.

Using the dataset from GSE32967 [37], we compared gene expression between small cell carcinoma (a subset of NEPC) and adenocarcinoma patient derived PCa xenografts. GSE Analysis showed that the androgen response hallmark was significantly decreased in the NEPC samples, while significant increases were observed in the serine metabolism and one carbon (tetrahydrofolate) metabolism pathways (Figure 5C). Several of serine and tetrahydrofolate leading-edge genes increased in our RNAseq analysis were also increased in the GSE32967 dataset (Figure 5D). Addition of CPT1A and AR gene expression to the heatmap showed that one NEPC sample had low AR as expected, but modest increased CPT1A associated with increased SHMT2, MTHFD2 and PSPH expression. Conversely, one sample with adenocarcinoma features showed less CPT1A associated with less SHMT2, MTHFD2, and PSAT1 expression (Figure 5D). We further investigated the direction of the relationship between CPT1A and AR expression in metastatic disease (Appendix A), using the Taylor et al. dataset [38]. A positive correlation was observed in the primary tumors, as expected from the role of androgens in regulating lipid metabolism [39]. However, this positive correlation was lost in the metastatic samples (R = −0.28, *p* = 0.25), where AR expression was significantly increased compared to primary tumors (Wilcoxon *p* = 1.1 × 10^−7^). The possibility that a strong anti-androgen blockade in these metastatic samples could reverse the correlation to less AR and more CPT1A expression, as it happens in NEPC, warrants further investigation.

### 2.6. Increased Lipid Catabolism and Decreased Androgen Response Is Associated with Poorer Progression-Free Survival

To further validate the role of CPT1A and lipid catabolism in advanced PCa, we turned to publicly accessible TCGA (The Cancer Genome Atlas) Firehose Legacy dataset (492 samples). Pathways were scored in each patient using GSEA and grouped according by median split of each pathway. Examination of the same GSEA pathways identified in our RNAseq data (Figure 2C) showed that the lipid catabolic process (Figure 6A), and the androgen response hallmark (Figure 6B), were significantly associated with progression free survival (PFS) but in opposite directions. The increase in lipid catabolism genes and the decrease in androgen response genes shortened the PFS. We did not observe significant PFS changes with the GO serine metabolism and the unfolded protein response hallmark pathways (Appendix A).

## 3. Discussion

This study reports on the connection between CPT1A overexpression (OE) and the metabolic and genetic consequences of its increased activity. Particularly, we found that CPT1A OE cells produced a significant number of acyl-carnitines that promote growth and resistance to stress insults, like excess lipids and androgen withdrawal. These later conditions are characteristic of advanced PCa, where androgen deprivation and excess circulating lipids frequently exists [40].

At the molecular level, this work provides insights into the metabolic changes precipitated by the excess use of lipid oxidation in the mitochondria, particularly the upregulation of the overall lipid catabolic process. As expected, not only CPT1A was increased, but also the ability to hydrolyze lipid stores via PNPLA2 (triglyceride lipase or ATGL) to increase the supply of fatty acids for the mitochondria (Figure 1B and Figure 2D). It is possible that new inhibitors for ATGL could be of therapeutic value for cancers with increased CPT1A activity [41]. Another molecular aspect of the increased CPT1A activity was the re-wiring of metabolism towards the serine biosynthesis pathway, which has been recently shown to be important in supporting mitochondrial function [30], and a driver in NEPC [42]. These unexpected amino acid metabolism changes are likely promoted by the strong activation of the adaptive ER Stress response, as indicated by the transcriptional upregulation of genes linked to the ER stress response, Figure 2. In fact, ATF4 is a key transcription factor translationally induced upon activation of the unfolded protein response or UPR [43]. This induction triggers an anti-stress response that promotes adaptation, or in the case of a chronic unresolved stress, it can promote cell death. Increased ATF4 regulates serine and glycine metabolism genes to drive de novo serine and glycine production, which can be used for antioxidant defense and glutathione production [44]. In the CPT1A-OE cells, the mitochondrial SHMT2 gene was significantly increased in OE versus KD cells, suggesting a strong induction of serine/glycine synthesis in the mitochondria. This is likely to provide antioxidant defense via the mitochondrial folate cycle that can generate NADPH [45], and carbon units to produce cystathionine, an intermediate in the synthesis of cysteine and ultimately glutathione (Figure 3).

The ER stress observed in the OE cells compared to the KD cells is likely a response to the high ROS production in the mitochondria and low SOD2 expression. This is a known mechanism to promote cancer growth [46]. This was an unexpected result considering the OE cells did not show signs of distressed or fragmented mitochondria. A possible explanation is the upregulation of the glutathione homeostasis pathway. We found that the glutathione-degrading and ER response gene CHAC1 (cation transport regulator homolog1) was increased in OE cells. This gene was discovered in a co-regulated group of genes enriched for components of the ATF4 pathway, including CCAAT/enhancer-binding protein beta (CEBPB), which also binds to the CHAC1 promoter [47,48]. All this evidence would suggest that increased expression of a glutathione-degrading enzyme and activation of the ER stress pathway will lead to cancer cell death. However, we did not observe such changes in the OE cells as the levels of glutathione did not change and cells were able to grow in the presence of exogenously added lipids (Figure 1 and Figure 4). Other studies in breast and ovarian cancer have shown that CHAC1 expression correlates with tumor differentiation and survival [49], suggesting that the observed ER stress in our models is likely stress-resolving and can promote disease progression. In fact, when we challenged the cells with commonly circulating fatty acids in the absence of androgens (a stressful environment), more SOD2 expression was observed in the OE cells and less on the KD cells. This underscores the increased antioxidant response capabilities of the OE cells and potential for survival and growth. This environment may promote adaptation of the OE cells to androgen deprivation, supporting progression to lethal disease. Recent studies have also shown that increased SOD2 activity can protect prostate cells when exposed to radiation [50].

How oxidative changes in the mitochondria connect with ER stress remains unknown. ER and oxidative stress have overlapping and intertwined functions in cancer [21]. Both promote epithelial mesenchymal transition, a key step of metastasis and tissue invasion of many tumor cells. In addition, detachment from the extracellular matrix activates the ATF4-HSPA5 branch of the UPR, which protects from anoikis by stimulating both autophagy and antioxidative stress responses [51]. As the CPT1A-OE cells prefer to grow in suspension [10], they might be using the ER stress response and the mitochondrial oxidative environment to transform to more aggressive tumors [52].

This study and the public databases provide evidence that lipid catabolism driven by CPT1A is associated with more aggressive disease. CPT1A-OE cells showed more SYP and ENO2 neuroendocrine marker expression compared to the KD cells. This suggests that in CRPC tumors, CPT1A activity can rewire metabolism to promote growth and transformation via activation of serine biosynthesis, folate cycle, and glutathione homeostasis, all geared to maintain an adequate redox balance in the cancer cells. The role of mitochondrial ROS in activating these pro-tumor antioxidant pathways warrants further investigation.

## 4. Materials and Methods

### 4.1. Cell Lines Fatty Acids and Drugs

LNCaP-C4-2 cells were purchased from the University of Texas MD Anderson Cancer Center (Houston, TX, USA). Cells were used at low passage number and grown in RPMI containing 10% FBS supplemented with amino acids and Insulin (Gibco, ThermoFisher, Walthman, MA, USA). Charcoal stripped serum (CSS) was used for androgen-deprived conditions. Lentiviral particles for shRNA and complete cDNA specific to CPT1A were prepared at the Functional Genomics facility at the University of Colorado. For transfection, the following shRNAs from the Sigma (St. Louis, MO, USA) library were used: For knockdown (KD), TRCN0000036279 (CPT1A-sh1, [17]) and control shRNA (NTshRNA) SHC202 were used. This specific shRNA has been used by us successfully in several models of cancer [10,15,16,17,52,53]. For CPT1A overexpression (OE), we used the ccsbBroad304-00359 clone from the CCSB-Broad lentiviral library as described [10]. Lentiviral transduction and selection were performed according to Sigma’s MISSION protocol. Puromycin (1 μg/mL) and blasticidin (5 μg/mL) from Sigma were used for KD and OE cell line drug selection, respectively. Fatty acids were purchased from Sigma, resuspended in ethanol for a stock solution of 10 mM and stored at −80 °C. Fatty acids were conjugated to albumin, (A7030, Sigma), before use at a 2 mM: 5% (Fatty acid:BSA) ratio in RPMI media. Etomoxir-HCL (CPT1 inhibitor) was purchased from Sigma and resuspended in PBS to 33.4 mM and stored at −20 °C.

### 4.2. Reverse-Transcriptase-PCR

For RT-PCR analysis, cDNA was synthesized (Applied Biosystems, Foster City, CA, USA) and quantified by real-time PCR using SYBR green (BioRad, Hercules, CA, USA) detection. Results were normalized to the housekeeping gene RPL13A mRNA and expressed as arbitrary units of 2^−ΔΔCT^ relative to the control group. Primer sequences:

RPL13A-F: 5-CCTGGAGGAGAAGAGGAAAGAGA.

RPL13A-R 5-TTGAGGACCTCTGTGTATTTGTCAA.

CPT1A-F: 5-TGGATCTGCTGTATATCCTTC.

CPT1A-R: 5-AATTGGTTTGATTTCCTCCC.

### 4.3. Western Blot Analysis

Protein extracts of 20 µg were separated on a 4–20% SDS-PAGE gel and transferred to nitrocellulose membranes as described [10]. Band signals were obtained and visualized with the LI-COR Biosciences system (Lincoln, NE, USA). Antibodies: CPT1A: 15184-1-AP, (Proteintech, Rosemont, IL, USA); GAPDH: CST 5174, (Cell Signaling Technology, Beverly, MA, USA); SOD2: CST 13141, Cell Signaling Technology.

### 4.4. Metabolomics and Acyl Carnitine Analysis

Cells were grown to 80% confluency before trypsinization and collection in 2 × 10^6^ aliquots. Samples were processed at Biological Mass Spectrometry Facility at the University of Colorado AMC (Aurora, CO, USA) using standard protocols. For the measurement of acyl-carnitines, samples were extracted in a solution of methanol, acetonitrile, and water (5:3:2) at a concentration of 1 million cells/mL in presence of acyl-carnitines, deuterated standards (NSK-B, Cambridge Isotope Laboratories, Tewksbury, MA, USA). Samples were analyzed via UHPLC-MS (Vanquish-Q Exactive, ThermoFisher, Walthman, MA, USA) as previously described [54]. Analysis of the most significant pathways was performed with the MetaboAnalyst web-based analytical program [55].

### 4.5. Intracellular Lipase Analysis

The enzymatic activity of intracellular lipase was measured using a substrate containing 3H Triolein (Perkin Elmer, Waltham, MA, USA) and human serum as a source of ApoC2 as described previously [56]. In brief, LNCaP-C4-2 cells overexpressing CPT1A (OE) and controls (EV) were grown to 80% confluence. Intracellular lipase was made accessible by lysing the cells in heparin containing M-PER cell lysis buffer (Pierce, Rockford, IL, USA). Intracellular lipase activity was determined by incubation with 3H Triolein substrate for 45 min at 37 °C. The protein concentration of the lysate was determined to calculate the Lipase-dependent hydrolysis (FFA release) of 3H Triolein per mg, per min.

### 4.6. Electron Paramagnetic Resonance Spectroscopy

Mitochondrial ROS production was measured by EPR using the mitochondrial-targeted spin probe 1-hydroxy-4-[2-triphenylphosphonio)-acetamido]-2,2,6,6-tetramethyl-piperidine,1-hydroxy-2,2,6,6-tetramethyl-4-[2-(triphenylphosphonio)acetamido] piperidinium dichloride (mito-TEMPO-H) as previously reported [57]. Cells were grown to 80% confluency prior to the EPR measurements. The mito-TEMPO-H probe was prepared in deoxygenated 50 mM phosphate buffer. Cells were washed and treated with mito-TEMPO-H 0.25 mM in Krebs-HEPES buffer (KHB) containing 100 μM of a metal chelator DTPA to avoid direct oxidation with metal ion or hydroxyl radical generation by Fenton reaction. Cells were incubated for 50 min at 37 °C, placed on ice, then gently scraped. 50 μL of cell suspension was loaded in an EPR capillary tube and EPR measurements were performed at room temperature using Bruker EMXnano X-band spectrometer [57]. EPR acquisition parameters were: microwave frequency = 9.6 GHz; center field = 3432 G; modulation amplitude = 2.0 G; sweep width = 80 G; microwave power = 19.9 mW; total number of scans = 10; sweep time = 12.11 s; and time constant = 20.48 ms. mito-TEMPO. Nitroxide radicals concentration was obtained by simulating the spectra using the SpinFit module incorporated in the Xenon software of the bench-top EMXnano EPR spectrometer followed by the SpinCount module (Bruker, Billerica, MA, USA).Total protein was extracted from analyzed samples and quantified with a Bio-Rad DC protein assay kit (Bio-Rad, Hercules, CA, USA), and nitroxide concentrations were normalized to total protein.

### 4.7. Statistics

Student *t*-tests or ANOVA tests were used to compare between groups, followed by post hoc tests when appropriate, alpha = 0.05. Analysis was carried out with GraphPad Prism software v8 (GraphPad Software, San Diego, CA, USA). All data represent mean ± SD, unless otherwise indicated.

### 4.8. RNAseq and Pathway Analysis

All cells (KD, OE, and their respective controls) were grown to 80% confluency before RNA isolation. RNA was extracted using a RNeasy Plus Mini Kit (Qiagen, Valencia, CA, USA). RNA quality was verified using a High Sensitivity ScreenTape Assay on the Tape Station 4200 (Agilent Technologies, Santa Clara, CA, USA) and measured with a Tecan Plate Reader (Thermo Fisher Scientific, Waltham, MA, USA). Library construction was performed using the Universal Plus mRNA Library Kit (NuGen Technologies, Redwood City, CA, USA), and sequencing was performed on the NovaSeq 6000 instrument (Illumina, San Diego, CA, USA) using paired-end sequencing (150 bp) by the University of Colorado Cancer Center Genomics and Microarray Core. Illumina adapters were removed using BBDuk (sourceforge.net/projects/bbmap) and reads <50 bp after trimming were discarded. Reads were aligned and quantified using STAR (2.6.0a) [58]) to the Ensembl human transcriptome (hg38.p12, release 96). Normalization and differential expression were calculated using the limma R package [29]. An interaction model within limma used to directly compare the OE and the KD. Gene set enrichment analysis was performed using the fGSEA R package (v1.10.0) with 10,000 permutations and the Hallmarks and GO Biological Processes gene set collections from the Molecular Signatures Database [59]. Heatmaps were generated with the ComplexHeatmap R package [60] following *z*-score transformation. RNA-sequencing data have been deposited into the NCBI Gene Expression Omnibus database (accession number GSE161243).

### 4.9. Public Database Analysis

RMA normalized prostate cancer microarray data was downloaded from GEO (GSE32967, [37]). Only the first replicate for each sample was used for the analysis to deal with uneven sample replicates. The most variable probe was selected to represent each gene. The limma R package was used to compare small cell carcinoma (*n* = 4) to adenocarcinoma (*n* = 3) prostate cancer. TPM normalized gene expression and clinical data from The Cancer Genome Atlas PRAD dataset (Firehose Legacy) [61], was downloaded from cBioPortal [62]. Gene Set Variation Analysis (GSVA) was performed using the GSVA R package [63], to score several gene sets with a Poison distribution. Patients were classified into high and low pathway groups by median score splitting. The survival (https://cran.r-project.org/web/packages/survival/citation.html) and survminer R packages (https://rpkgs.datanovia.com/survminer/index.html) were used to generate Kaplan-Meier curves and perform log rank tests. For the CPT1A and AR gene expression correlations, Log2 normalized whole transcript mRNA expression values from Taylor et al. [38] prostate cancer samples were downloaded from cBioPortal [62]. Expression of AR and CPT1A between primary and metastatic samples were compared using a Wilcoxon rank sum test. Pearson correlation was calculated for the correlation between AR and CPT1A expression in either primary or metastatic tumors.

## 5. Conclusions

The overall goal of this study is to understand the lipid metabolic underpinnings of advanced prostate cancer so that metabolic therapies can be designed effectively. Overall, we provide evidence that CPT1A activity may have a relevant role in advanced PCa, including transformation to NEPC. Etomoxir is a potent inhibitor of CPT1 that has been used in clinical trials in Europe [64], but it is not currently being used in the USA. Considering all the toxic chemotherapeutic agents used in cancer treatments, the potential for drugs like etomoxir to impact cancer growth and drug response warrants investigation.

## Figures and Tables

**Figure 1 cancers-12-03431-f001:**
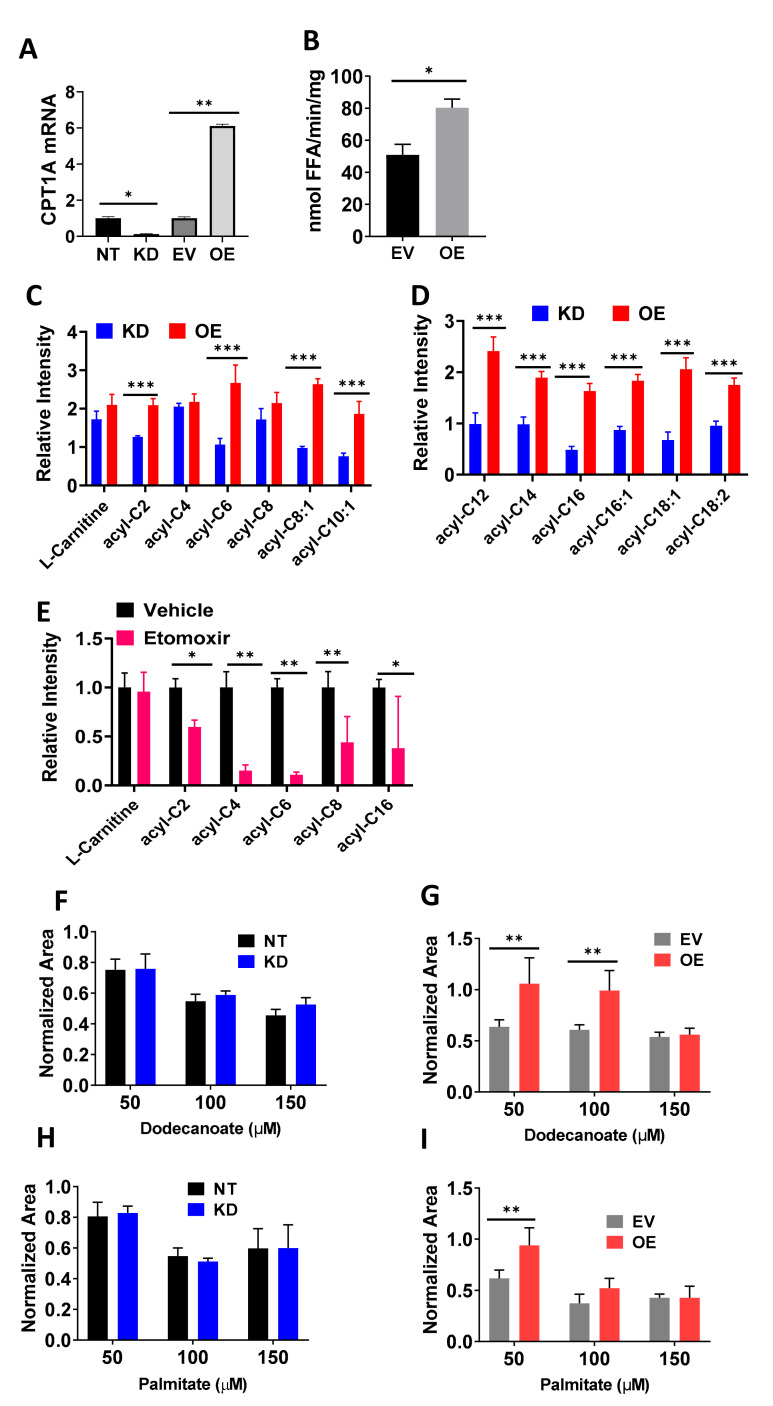
Cells with overexpression of carnitine palmitoyl transferase I (CPT1A) show a lipid catabolic phenotype and increased growth when supplemented with fatty acids. (**A**) mRNA expression of C4-2 cells with decreased (knockdown (KD)) and overexpression (OE) of CPT1A cells and their respective controls (non-targeting (NT) and empty virus (EV)). (**B**) Intracellular triglyceride lipase assay in OE cells. (**C**,**D**) Short and medium (**C**) and long chain (**D**) acyl carnitine content in OE (red) versus KD (blue), normalized to their own controls. (**E**) Acyl carnitine changes in C4-2 parental cells treated with etomoxir (100 μM) for 48 h. (**F**,**G**) Colony growth after treatment with C12:0 fatty acid conjugated to BSA for 14 days and normalized to BSA-only treatment. (**H**,**I**) Colony growth after treatment with C16:0 fatty acid conjugated to BSA for 14 days and normalized to BSA-only treatment. * *p* < 0.05, ** *p* < 0.01, *** *p* <0.001.

**Figure 2 cancers-12-03431-f002:**
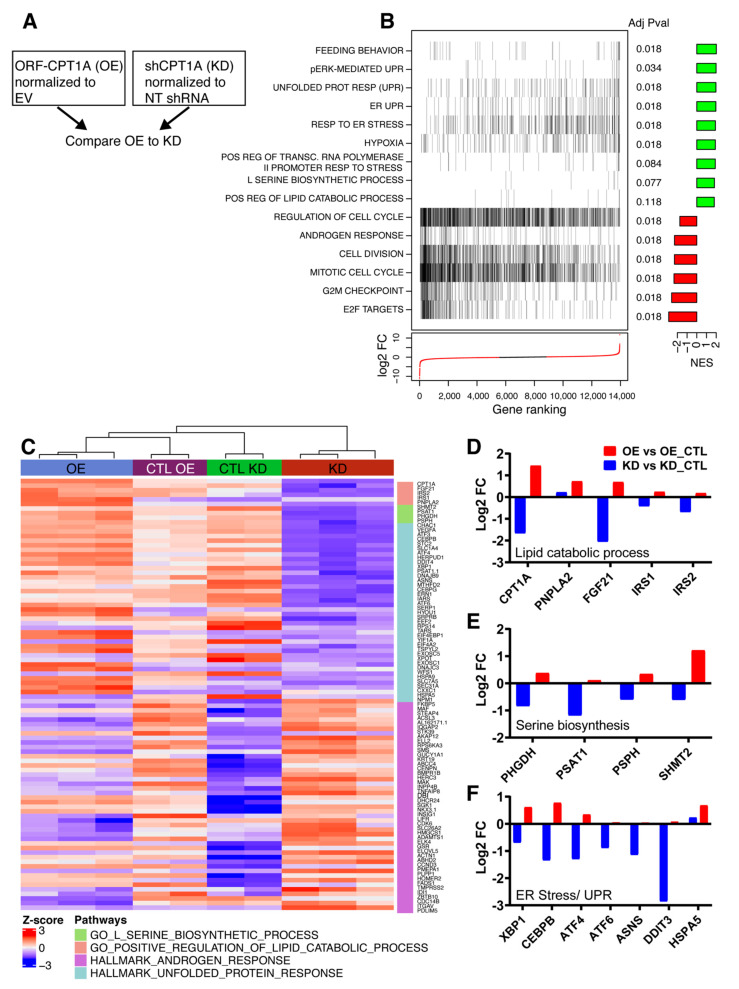
RNAseq analysis of CPT1A OE cells shows increased endoplasmic reticulum (ER) stress response, serine metabolism, and less androgen receptor (AR) signaling. (**A**) Schematic of the RNAseq analysis paradigm with the CPT1A-KD and OE cells. (**B**) Gene set enrichment analysis was performed on the fold change of the comparison. Normalized Enrichment Scores (NES) and False discovery rate (FDR) adjusted *p*-values are shown for select pathways, as well as the rank of those genes in fold change ranking, are plotted. (**C**) Heatmap of the leading-edge genes for select pathways. (**D**–**F**) Gene expression graphs of significant genes associated with lipid catabolism (**D**), serine biosynthesis (**E**) and ER stress (**F**) in OE versus KD comparison after adjustment to their respective control cell lines. Adj *p* value < 0.001.

**Figure 3 cancers-12-03431-f003:**
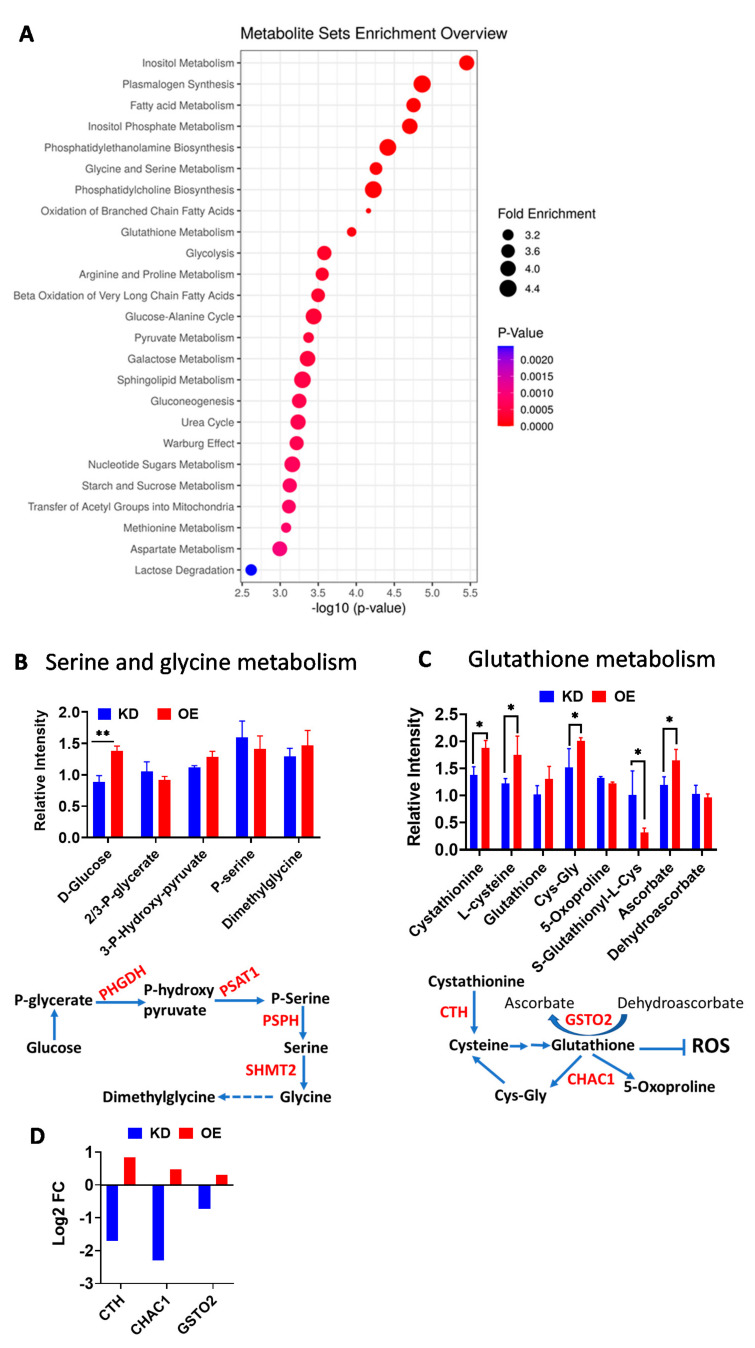
Excess CPT1A is associated with serine and glycine metabolism pathway and glutathione homeostasis metabolites. (**A**) Metabolite set enrichment analysis of OE metabolites versus KD cell metabolites after normalization to their own controls. The *p*-value is defined by the color scale, and the enrichment by the size of the circle. (**B**) Relative abundance of metabolites associated with the serine and glycine biosynthesis via the SHMT2 gene in the mitochondria. A scheme of the pathway is shown under the graphs, highlighting the sequence of events from glucose to synthesis of serine, glycine and the one carbon metabolism compound dimethylglycine. The genes significantly altered in OE cells from the RNAseq analysis involved in the pathway are shown in red. (**C**) Relative abundance of metabolites associated glutathione (GSH) homeostasis. The S-glutathionyl-L-Cys metabolite is associated with ineffective handling of oxidative stress. A scheme to the pathway is shown below. * *p* < 0.05, ** *p* < 0.01. (**D**) Gene expression analysis of OE versus KD cells for CTH (*p* = 6.29 × 10^−13^), CHAC1 (*p* = 3.6 × 10^−12^) and GSTO2 (*p* = 0.003) genes (adj *p*-value).

**Figure 4 cancers-12-03431-f004:**
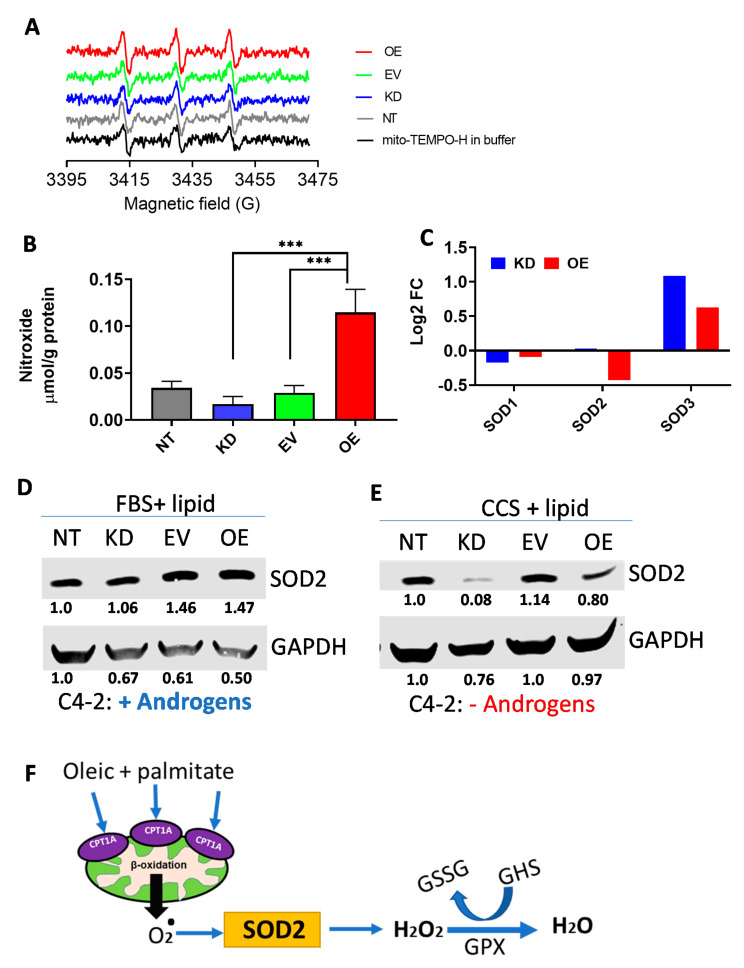
Mitochondrial reactive oxygen species (ROS) are increased in CPT1A-OE cells. (**A**) Electron paramagnetic resonance spectroscopy (EPR) traces of the four cell lines tested, including the mitochondrial probe by itself. Amplitude and linewidth provide information of the amount of ROS. The concentration was acquired by Spin Fit followed by SpinCount module (Bruker). (**B**) Quantification of the EPR traces normalized to protein content. *** *p* < 0.001. (**C**) Gene expression analysis of OE versus KD cells for SOD1 ns), SOD2 (*p* = 6.9 × 10^−4^), and SOD3 (*p* = 0.015). Only SOD2 is a mitochondrial enzyme. (**D**,**E**) Western blots of SOD2 expression after incubation with fatty acids (Oleic and palmitate mixture; 25 µM each) in FBS (**D**) or charcoal stripped serum (CSS) (**E**) for 48 h. (**F**) Schematic of the role of SOD2 in metabolizing superoxide and the role of glutathione in eliminating H_2_O_2_. GPX = Glutathione peroxide. GSSG = oxidized glutathione.

**Figure 5 cancers-12-03431-f005:**
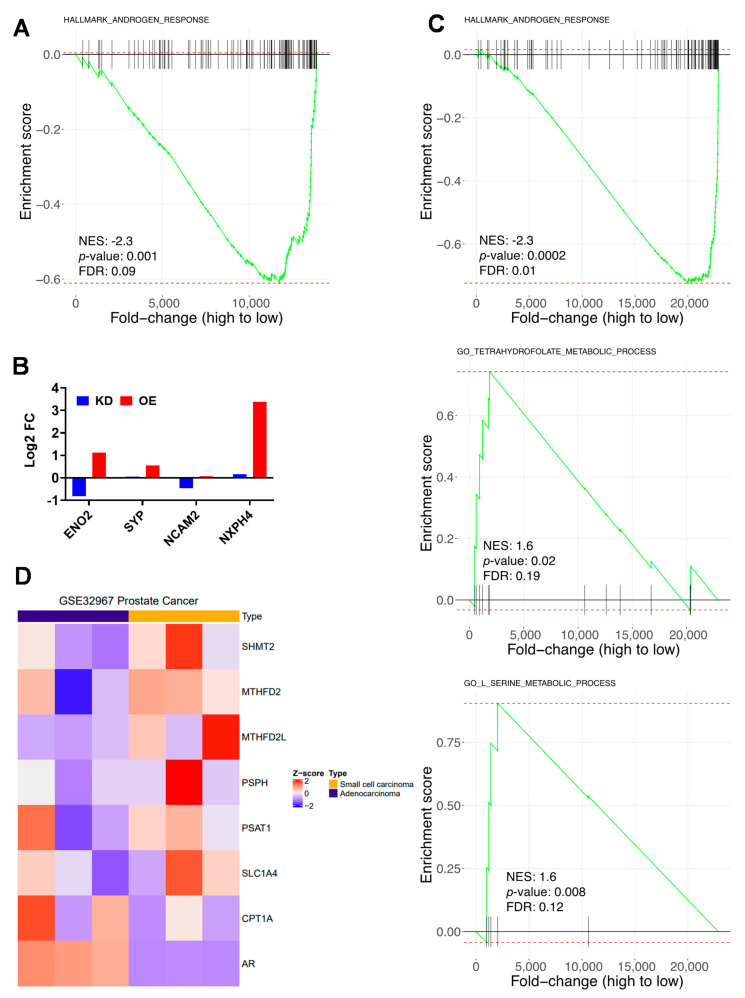
Lipid and serine metabolism genes are associated with less androgen signaling and a neuroendocrine phenotype. (**A**) Gene set enrichment analysis (GSEA) plot of the OE versus KD analysis showing the decrease in the androgen response hallmark in OE cells. (**B**) Gene expression analysis of OE versus KD cells for neuronal-like markers like ENO2 *(p =* 6.1 × 10^−10^*),* SYP (*p* = ns), NCAM2 (*p* = 4.14 × 10^−5^), NXPH4 (*p =* 1.6 × 10^−17^). (**C**) GSEA plots of the public GSE32967 dataset showing the comparison between small cell carcinoma (*n* = 3) versus adenocarcinoma (*n* = 3) patient xenografts for the Hallmark Androgen Response; GO_Serine and GO_Tetrahydrofolate pathways. Normalized enrichment scores (NES) and statistical significance are indicated in the plots. (**D**) Heatmap of the leading-edge genes from the serine and tetrahydrofolate functional GSEA plots shown in panel C, which are also genes increased in the OE cells compared to KD cells. CPT1A and AR are also included and show an inverse correlation in one of the samples of each subtype of PCa studied.

**Figure 6 cancers-12-03431-f006:**
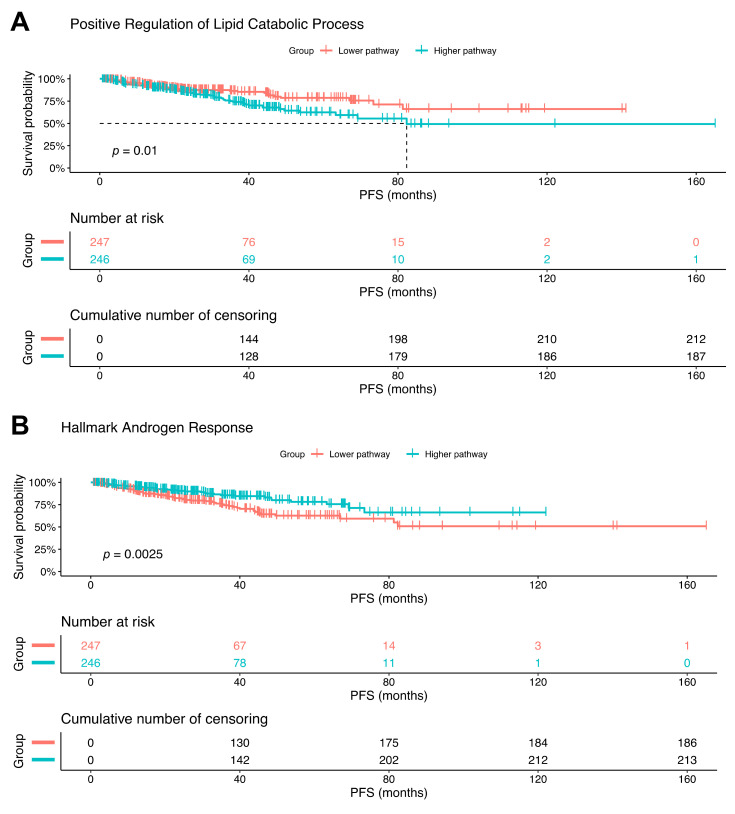
Increased lipid catabolism and decreased androgen response is associated with less progression-free survival. The TCGA prostate adenocarcinoma (PRAD) Firehose legacy dataset (*n* = 492) was divided by median split into high and low pathway scores. Pathways from our OE versus KD RNAseq analyses (Figure 2C) were then studied for progression free survival (PFS) in the TCGA PRAD dataset. Kaplan–Meier (KM) plots for Lipid Catabolism Process (**A**) and Androgen Response Hallmark (**B**), with the number at risk and *p*-value (logrank test) are shown.

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
