# Peer review of "CPT1A Over-Expression Increases Reactive Oxygen Species in the Mitochondria and Promotes Antioxidant Defenses in Prostate Cancer"

_cancers, 2020, doi:10.3390/cancers12113431_

Round 1
Reviewer 1 Report
To study the lipid metabolism in prostate cancer, the authors examined the consequences of altered expression of the mitochondrial enzyme CPT1A in LNCaP C42 cells using transcriptomic and metabolomic analysis. The authors suggested upregulated ER stress, serine biosynthesis and lipid catabolism and downregulated androgen response in in the overexpressed cells. Also, the authors claim that SOD2 is increased in the OE cells for better antioxidant defenses.
General points:
- The introduction section can be more succinct
- Since majority of the data is based on one cell line LNCaP-C42. Can the authors support the results by showing the expression of CPT1A expression in human prostate cancer use the available clinical datasets, and see if it has a negative correlation with androgen receptor/ar signaling, and positive correlation with serine/lipid metabolism
Specific points:
Figure 1B is extremely similar to a figure in author's previous paper (cells, 2019 figure 1B). Please change.
Figure 2B, please enlarge the font.
Figure 2C the huge differences between ctrl-oe and ctrl-kd is concerning, can you present the heatmap of normalized fold change of the indicated genes.
Figure 2E, there is a typo on PHGDH
Figure 3. Please be consistent in presenting the data by comparing KD and OE, which I believe it is true in figure 3a. However, only d-glucose showed a significant increase in OE compared to KD, other metabolites do not show significant changes. .But the authors decided to compare oe to ev in figure 3b. Then in figure 3c the authors go back to compare oe to kd. Please be consistent.
In Figure 4: To compare if SOD2 response to lipid stimulation, the samples in figure 4 d (FBS wo lipids) and e (FBS with lipids) should be processed at the same time and ran in the same gel side by side. To compare the effect of androgen on SOD2, the authors should compare cells cultured in css versus cells cultured in css + DHT. And blot for PSA as a positive control. This experiment will also provide evidence for the statement of decreasing androgen response genes in OE cells (figure 5a).
In figure 5: is there a CPT1A upregulation and increased lipid metabolism in the GSE32967 dataset comparing small cell carcinoma versus adenocarcinoma patient xenografts?
Author Response
Thank you for the valuable comments.
Please see attached PDF with the responses to the 3 reviewers

Reviewer 2 Report
The manuscript submitted by Isabel Schlaepfer and co-authors aimed to address the role of CPT1A in prostate cancer by comparing cells lines with enzyme knockdown or overexpression. The manuscript is interesting, well written and organized.
Comments:
- More detail is needed in M&M concerning the fatty acids used in different experiments. Only in figure legends this is information is provided. Authors also should explain why they use different approaches, for example, Dodecanoate in figure 1 and oleic and palmitate mixture in figure 4. Also, a general overview of cell treatments is needed.
- In M&M it is indicated that “Etomoxir-HCL (CPT1 inhibitor) was purchased from Sigma and resuspended in PBS to 30 uM”, but in figure 1’ legend the concentration indicated is 100 uM. Please clarify. Again, a general explanation of the experimental strategy and cell treatments with different compounds and drugs, and their respective concentrations would be helpful. Authors also should explain why this so high etomoxir concentration was selected?
- Does etomoxir is being used in clinical trials? What is the possibility of inhibiting CPT1A in patients in the near future. Can authors discuss this, please?
- Considering the RNASeq analysis 1157 genes were overexpressed and 1385 downregulated comparing CPT1A OE against KD. Authors gave tittle attention to this fact, focusing almost exclusively in the discussion of upregulated genes. Any comment? Could be downregulated pathways be physiological significant?
- In general, figures are too small and very hard, or even impossible, to read.
Minor comments:
Line 104: should be hormone-dependent cancers instead of “hormonal cancer”.
Author Response

(The authors gave the same response as above.)

Reviewer 3 Report
Joshi and co-authors investigated the implication of the mitochondrial enzyme CPT1A and its levels of expression in prostate cancer progression. Overall, this research is a well-designed and performed study, and the content of this manuscript is of major interest. Nevertheless, the following issues need to be addressed:
- The results obtained in this study was obtained using only a prostate cancer cell line as in vitro model (LNCaP-C4-2). In my opinion, the study should have been extended to other PCa cell lines. Besides, this is a Cancers requirement as stated in the “Instruction for the authors” of this journal: ….”research articles using only one cell line for the experiments will not be considered for publication (experiments need to be repeated on 1-2 more cell lines)”. The authors should, at least, explain the reason(s) that led them to use only this cell line.
- The authors used 1 shRNA to knockdown CPT1A. In my opinion this is not enough because for gene silencing experiments should be used at least two gene-specific shRNAs.
- The authors should provide, as supplementary material, the whole uncropped Western blots of figure 1B, 4D, 4E, 4F, showing all bands and molecular weight markers.
- The resolution of Figures 2B, 2C, and 3A is vey low. For the reader is very hard to see the names of the genes and the other words.
Minors
- In the main text, “Figure x” should be cited as “(Figure x)”
- Please check the whole manuscript for typos and spaces between words
- line 387: “The enzymatic activity of intracellular lipase activity”. Maybe the second “activity” should be deleted
Author Response

(The authors gave the same response as above.)

Round 2
Reviewer 1 Report
Thank the authors for their detailed reply and careful revision. The authors addressed all of my questions.
Reviewer 3 Report
The authors have addressed all my comments/suggestions. I found their responses quite satisfactory and the revised version has been much improved. I now recommend the paper for publication in Cancers